# Rapid exchange cooling with trapped ions

Spencer D. Fallek [1] ✉, Vikram S. Sandhu [1], Ryan A. McGill [1], John M. Gray[1], Holly N. Tinkey[1], Craig R. Clark[1] & Kenton R. Brown [1]

The trapped-ion quantum charge-coupled device (QCCD) architecture is a leading candidate for advanced quantum information processing. In current QCCD implementations, imperfect ion transport and anomalous heating can excite ion motion during a calculation. To counteract this, intermediate cooling is necessary to maintain high-fidelity gate performance. Cooling the computational ions sympathetically with ions of another species, a commonly employed strategy, creates a significant runtime bottleneck. Here, we demonstrate a different approach we call exchange cooling. Unlike sympathetic cooling, exchange cooling does not require trapping two different atomic species. The protocol introduces a bank of "coolant" ions which are repeatedly laser cooled. A computational ion can then be cooled by transporting a coolant ion into its proximity. We test this concept experimentally with two $^{40}\text{Ca}^+$ ions, executing the necessary transport in 107 $\mu$s, an order of magnitude faster than typical sympathetic cooling durations. We remove over 96%, and as many as 102(5) quanta, of axial motional energy from the computational ion. We verify that re-cooling the coolant ion does not decohere the computational ion. This approach validates the feasibility of a single-species QCCD processor, capable of fast quantum simulation and computation.

Existing quantum computers are limited to tens or hundreds of qubits[1–3]. To build larger systems, the primary challenge is to increase qubit count without sacrificing operational fidelity. The QCCD architecture is one vision to expand systems based on trapped-ion qubits[4]. This architecture employs a modular approach, where gate and measurement operations are executed locally on small ion crystals. Ion transport is used to rearrange the crystals between gate operations. Fast transport is performed in microfabricated ion traps by applying time-varying voltages to multiple control electrodes[5–8]. Recent experiments have integrated transport and gate operations together to execute fault-tolerant logical operations, thereby substantiating the promise of the QCCD scheme[9–11].

In many architectures, including the QCCD approach, two-qubit gates represent the most challenging operation to perform with high fidelity. Leading two-qubit gate approaches, such as the light-shift gate[12] and the Mølmer-Sørensen gate[13], require cold ions that are well within the Lamb-Dicke regime for high-fidelity performance. Electric-field noise and ion transport can excite ion motion in unintended ways, making it challenging to maintain ions in the Lamb-Dicke regime throughout the course of an algorithm. Sympathetic cooling has been

employed to address this problem[14]. Here, each ion used in the computation is co-trapped with another ion of a second species. The second species is laser cooled, and through their shared motional modes, the computational ion is cooled without perturbing its internal state. This approach has enabled longer circuit depths, but with a cost in experiment duration and complexity. Regarding duration, laser cooling of a mixed-species crystal may take a few milliseconds even for only a single shared mode[15–17]. Consequently, cooling often dominates algorithm runtime, for example consuming as much as 68% of the total duration in Refs. 2,18. Regarding experiment complexity, trapping a second species requires an additional set of laser sources, each of which must be focused onto the ion crystals. This greatly increases the density of optical elements, regardless of whether bulk optics or integrated optics are employed[19–21]. Lastly, transporting mixed-species crystals quickly[22], or through trap junctions[23], presents important challenges when compared to the single-species case.

In this work, we explore exchange cooling, an alternative to sympathetic cooling[24,25]. The protocol introduces a bank of coolant ions, stored in a separate region of the trap, where they are continually laser cooled. To cool a computational ion, a single coolant ion is removed

[1]Georgia Tech Research Institute, Atlanta 30332 GA, USA. ✉e-mail: spencer.fallek@gtri.gatech.edu

from the bank and brought into close proximity to a hotter computational ion. The ions are held at separation $d$ for duration $t_{ex}$ during which they exchange motional energy via their mutual Coulomb interaction. The rate of exchange $\Omega_{ex}$ is proportional to $d^{-3}$, so it is desirable to make this separation as small as possible to maximize the coupling[26]:

$$\Omega_{ex} = \frac{q^2}{4\pi\epsilon_0 d^3 \sqrt{m_1 m_2} \sqrt{\omega_1 \omega_2}}. \tag{1}$$

Here, we specify the rate of exchange for axial motional energy, where $q$ is the electron charge, $m_1$ and $m_2$ are the ion masses and $\omega_1$ and $\omega_2$ are the individual ion axial mode frequencies absent the Coulomb coupling. Duration $t_{ex}$ is calibrated such that the two ions fully swap their energies. After the exchange, the coolant ion is returned to the bank for re-cooling, while the computational ion is transported to its next position in the QCCD algorithm.

Exchange cooling addresses the primary issues with sympathetic cooling: long runtimes and experimental complexity. Re-cooling of the coolant ions can occur while operations are being executed on the computational ions. This parallelization alleviates the large runtime penalty associated with laser cooling. The coolant ions can be of the same atomic species as the computational ions, thereby removing the experimental challenges discussed above. Recently, the *omg* blueprint was proposed to execute sympathetic cooling by employing different internal states of a single-species crystal[27]. This scheme does not allow for parallel laser cooling during quantum operations and requires additional interconversion pulses to store ground-state computational qubits during cooling. Again, exchange cooling may prove to be a faster and simpler alternative.

## Results

Motional energy exchange between a pair of ions held in separate, static potential wells has been demonstrated in Refs. 26,28. Our work integrates the necessary ion transport (dynamic operation), shown in Fig. 1. To utilize exchange cooling in an algorithm, the ions must be separated far enough to allow for re-cooling of the coolant ion without incurring errors on the computational ion. In our experiment, we confine two $^{40}Ca^+$ ions using a Sandia National Laboratories Peregrine trap held at 4.5 K[29]. The system incorporates dedicated laser beams, one for each ion, to address the $S_{1/2} - P_{1/2}$ transition (397 nm, used for Doppler cooling and

state detection) and $S_{1/2} - D_{5/2}$ transition (729 nm, used for sideband cooling and Ramsey pulses). Laser beams to depopulate the $D_{3/2}$ states (866 nm, used for repumping) and $D_{5/2}$ states (854 nm, used for deshelving) address both ions simultaneously[30]. Prior to exchange transport, the ions are confined in harmonic wells separated by 140 μm. As discussed below, the initial 140 μm separation is sufficient to allow for sideband cooling of the coolant ion without crosstalk to the computational ion. For an energy exchange, we implement a control voltage waveform to transport the ions into a double-well potential with nominal ion separation $d_{in}$ of 14 μm.

Our strategy with the exchange cooling process is to limit exchange coupling as the ions approach one another and as they are split apart. Coupling is suppressed by keeping the ion mode frequencies off-resonance with nonzero detuning $\delta\omega = (\omega_1 - \omega_2)$. Then we can optimize exchange transport in three parts, as outlined in Fig. 1: *a)* develop a waveform to bring the ions to separation near $d_{in}$, *b)* determine the compensating potentials and the duration $t_{ex}$ which achieve a full energy exchange, and *c)* develop a waveform to split the ions and return them to their starting positions. We insert delays after each part to allow our filtered waveform potentials to settle (steps $s_a$–$_c$). A detailed description of the optimized transport is provided in Transport Sequence. Here, we highlight our calibration procedure for part *b*, the key portion of the exchange transport.

Part *b* is executed in multiple steps, manipulating both $A_{ex}$, the maximal fraction of energy exchanged between the two ions, and $t_{ex}$, the interval allowed for exchange. As shown in the inset of Fig. 1, we seek to bring $A_{ex}$ from zero to unity in step $b_1$, hold for $t_{ex}$ in step $b_2$, and then return $A_{ex}$ to zero in step $b_3$. This is achieved through fine manipulation of the double-well potential. Manipulations must consider stray fields which cause deviation from design (see Double-well Modeling). For small oscillations of the ions about their equilibrium positions, when the potential can be approximated to second order in the ion displacements[26], $A_{ex}$ is given by:

$$A_{ex} = \left(1 + \frac{\delta\omega^2}{4\Omega_{ex}^2}\right)^{-1}. \tag{2}$$

To modify $A_{ex}$, we make first- and second-order adjustments to the double-well potential using two sets of compensating potentials. Each

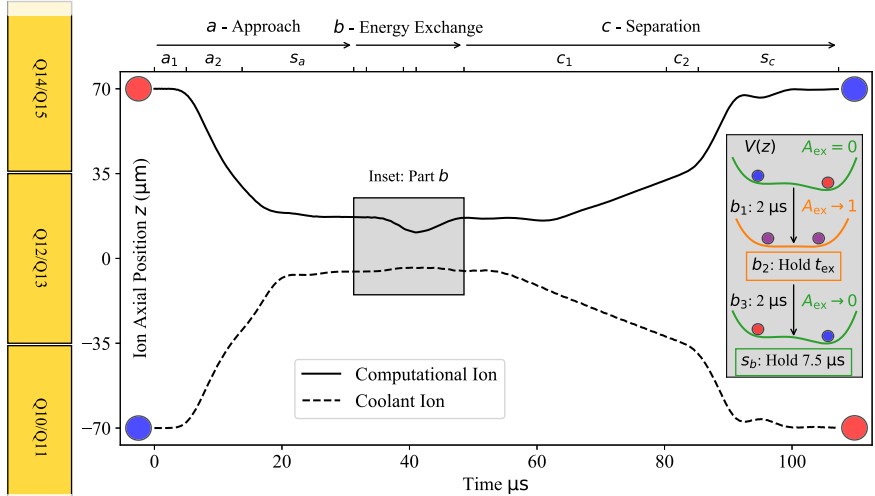

**Fig. 1 | Exchange cooling in the Peregrine trap.** Simulated axial trajectories of the computational ion and coolant ion through exchange-cooling transport. Positions are given relative to the center of the Q12/Q13 Peregrine trap electrodes. Electrodes are shown schematically in gold on the left. Axial motional energy transfers from an initially hot (red) ion to a cold (blue) ion. Top axis: Exchange transport is separated into three parts *a*, *b*, and *c*, which are themselves divided into smaller steps (see main text and Table 1). Inset: Modifications of the axial double-well potential $V(z)$ through part *b* (not to scale). The initial off resonance double-well potential (green) is interpolated onto resonance to generate the exchange potential (orange), at which point the ions exchange energy (purple).

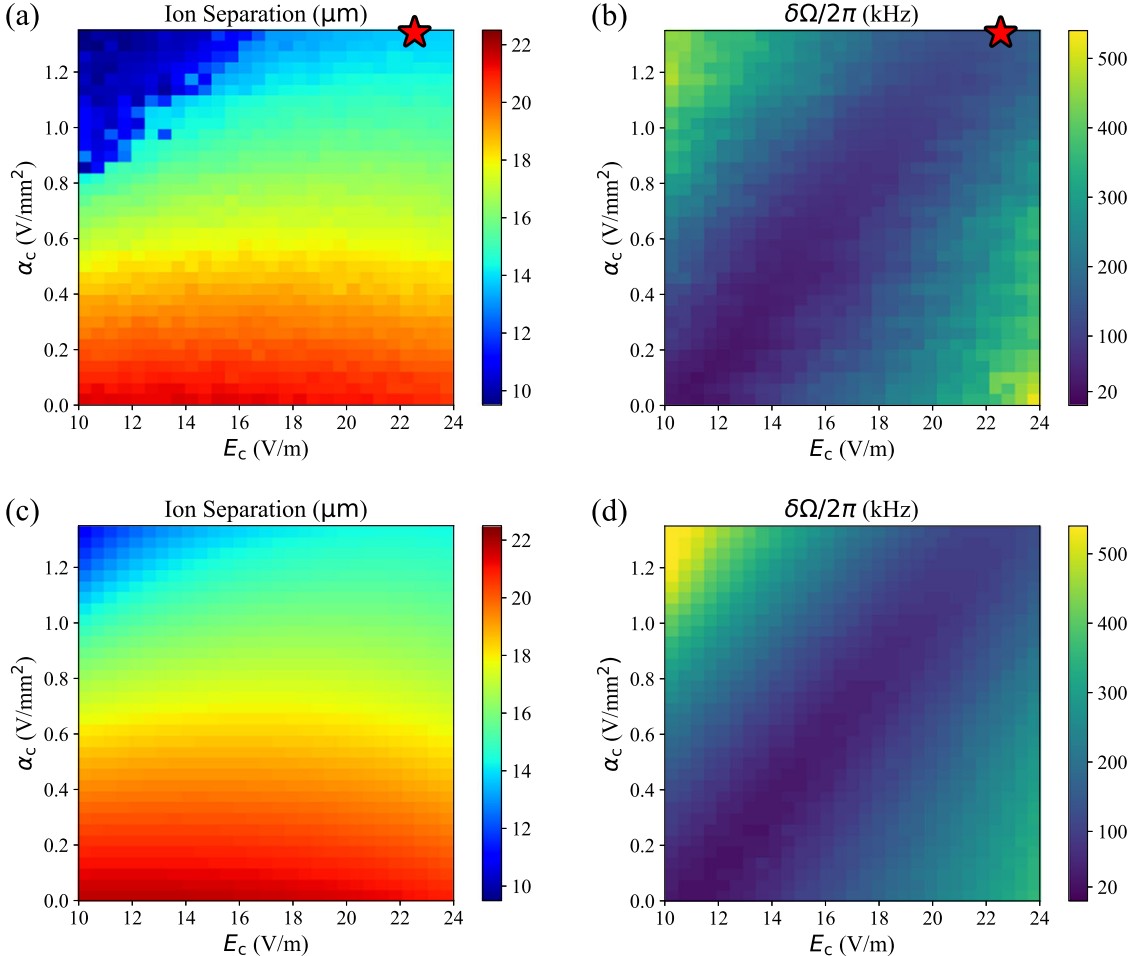

**Fig. 2 | Double-well and exchange-coupling characterization.** Measurements of ion separation (**a**) and $\delta\Omega$ (**b**) when varying the linear and harmonic compensating potentials. (**c**) and (**d**) are corresponding simulations (see Ion Motional Simulations). The red star in (**a**) and (**b**) indicates the value of compensating potentials used for a resonant exchange.

adjustment is designed to generate a specific electric potential $V(z)$ along the trap axis $z$. The first compensating potential set approximates a constant electric field ($V(z) = E_c z$). The second compensating potential set approximates a harmonic well ($V(z) = \alpha_c z^2$), with its minimum at the designed center of the double-well. These compensating potentials modify $d$ and $\delta\omega$, providing control over the parameters necessary to adjust $A_{ex}$.

In Fig. 2, we quantify this dependence by varying $E_c$ and $\alpha_c$ while the ions are held in an otherwise static double-well potential. We plot the ion separation $d$ as determined from images acquired with an EMCCD camera. The Coulomb interaction between the ions means that $\omega_1$ and $\omega_2$ are not eigenmodes of the axial motion, and we cannot measure $\delta\omega$ directly. Rather, we measure the difference between eigenmodes of the coupled system, $\delta\Omega = (\Omega_+ - \Omega_-) = \sqrt{4\Omega_{ex}^2 + \delta\omega^2}$. Along the diagonal blue stripe in Fig. 2b, $\delta\Omega$ reaches a minimum, and the ions are on resonance with $\delta\omega = 0$[26]. Here, $A_{ex} = 1$, and a full energy exchange is possible. In experiment, for step $b_2$, we choose compensations which adjust $d_{in}$ to its designed value and bring the ions into resonance: ($\alpha_c = 1.33$ V/mm², $E_c = 23$ V/m, $A_{ex} = 1$). We calibrate the resonant $E_c$ roughly once per day to optimize the energy transfer in Fig. 3a and track drifts of a few V/m. For settling steps $s_a$ and $s_b$, we desire $A_{ex}$ near zero: ($\alpha_c = -0.16$ V/mm², $E_c = 24$ V/m, $A_{ex} < 0.5\%$). To engage (disengage) an energy exchange in step $b_1$ ($b_3$), we ramp the necessary compensating potentials linearly over a 2 $\mu$s interval.

Next, we determine the optimal duration $t_{ex}$ to complete the calibration of part $b$. For this, we sideband cool the axial mode of the coolant ion near the ground state and Doppler cool the computational ion to ~ 15 quanta. We execute the full exchange transport and vary the delay $t_{ex}$ during part $b_2$ while the ions are on resonance. Figure 3a gives a plot of the mean energy of each ion as $t_{ex}$ is varied showing the expected oscillations; also shown are the results of a simulation based upon experimentally calibrated parameters. Discrepancies in the oscillation amplitude between simulated and measured data can be explained by heating as discussed below. Due to the finite response time of our electrode filters, the potential is not constant during the exchange, leading to the deviations from sinusoidal behavior seen both in simulations and in data. For the experiments below, we set $t_{ex} = 5.8$ $\mu$s, leading to a total roundtrip transport duration of 107.3 $\mu$s.

To measure the cooling efficiency of the optimized process, we vary the initial temperature of the computational ion prior to exchange transport. Here, we increase the ion temperature beyond the usual Doppler limit by illuminating the ion with a blue-detuned 397 nm laser beam. As shown in Fig. 3b, the dynamic exchange process is effective regardless of the initial thermal state, reducing the computational ion temperature back into the Lamb Dicke regime. In the hottest tested case, we reduce the computational ion temperature from 106(5) quanta to 3.9(2) quanta. The slight upward slope in the data reveals the efficiency of the energy transfer, where a linear fit shows that 98.1(1)% of the initial mean energy is removed

(a)

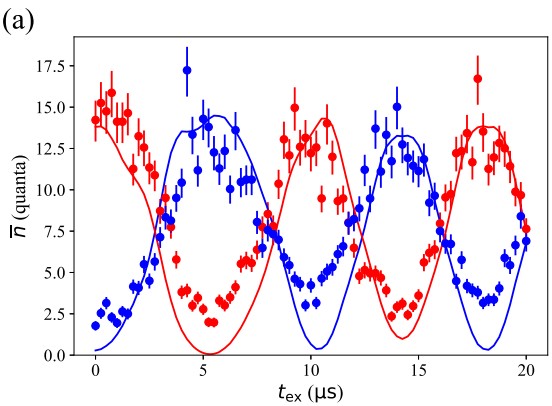

(b)

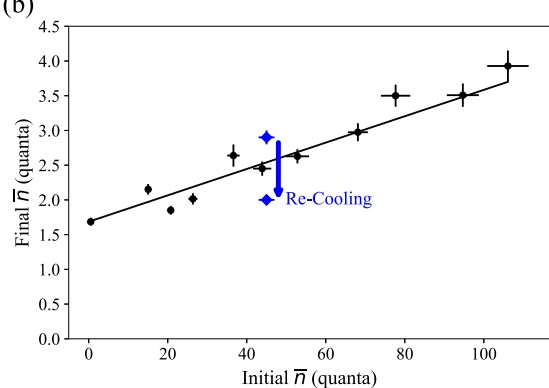

**Fig. 3 | Exchange cooling calibration and performance. (a)** Oscillations in axial mode temperature $\overline{n}$ following exchange transport as a function of $t_{ex}$. Energy swaps between the computational ion (red) and the coolant ion (blue). Our procedure for determining $\overline{n}$ and its uncertainty using a fixed-length sideband pulse is described in Temperature Measurements. Solid lines are simulations, as described in Ion Motional Simulations. **(b)** Computational ion temperature after exchange transport as the initial temperature is varied, along with a linear fit (solid black line). The blue arrow indicates the results of an experiment in which the coolant ion was re-cooled after the first exchange and a second exchange was attempted. Horizontal (vertical) error bars represent the uncertainty in ion temperature before (after) exchange transport measured using the sideband flopping method.

from the computational ion. The small remaining inefficiency is likely caused by an inability to set $E_c$ more precisely due to the limitations of our hardware (see Ion Motional Simulations). We can repeat the exchange process to achieve even lower temperatures as indicated by the blue arrow in Fig. 3b. Here, we perform a first exchange which reduces computational ion temperature from 45(2) quanta to 2.9(1). Then we re-cool the coolant ion via sideband cooling and execute a second exchange, achieving a final temperature of 2.00(7) quanta.

Besides the residual energy which is not exchanged, both ions acquire some additional energy from baseline heating during the transport. In the leftmost point of Fig. 3b, the computational ion is initially sideband cooled to a temperature near the ground state. Its temperature rises to 1.69(6) quanta due to the transport, which represents the lowest achievable temperature in our protocol. Such heating has been studied in the context of ion merge operations, where the duration spent at lower axial mode frequencies is a key parameter due to the increased effects of anomalous heating there[6,31]. Correspondingly, it is advantageous to execute step $s_a$ and part $b$ as quickly as possible, where axial frequencies drop into the range of $400-600$ kHz (see Transport Control).

Using exchange cooling in an algorithm requires that neither exchange transport nor re-cooling of the coolant ion decohere the computational ion. As depicted in Fig. 4, we verify this by performing a Ramsey experiment on the computational ion. The ions begin at a separation of 140 μm, around ten times larger than the beam waists of our 729 nm laser beams. The computational ion is sideband cooled near the ground state, while the coolant ion is merely Doppler cooled. A composite $\pi/2$ pulse prepares the computational ion in an equal superposition of $S_{1/2}$ Zeeman states. With the computational ion in the superposition state, sideband cooling brings the coolant ion near the ground state in 909 μs. This is followed by exchange transport, a second composite $\pi/2$ pulse for analysis, and detection of the computational ion's state. To alleviate the effect of slow magnetic field drifts, a spin-echo composite $\pi$ pulse is employed on the computational ion midway between preparation and analysis $\pi/2$ pulses (see Ramsey Sequence).

Figure 4b gives the results of this experiment as we vary the relative phase between the preparation and analysis $\pi/2$ pulses; the Ramsey fringe contrast is 96.0(7)%. Repeating the experiment without re-cooling and transport, but holding the ions stationary for an equivalent duration, yields a contrast of 96.4(6)%. Hence, both re-cooling and exchange transport do not adversely decohere the

internal state of the computational ion. We attribute the difference in phase between the two scenarios in Fig. 4b to magnetic field gradients which shift the qubit frequency as the ion undergoes exchange transport[32]. In an algorithm, this phase can be calibrated and corrected in subsequent pulses.

## Discussion

The performance demonstrated in this proof-of-principle realization is already sufficient for integration into existing QCCD architectures. The protocol utilizes only one species of ion to execute cooling an order of magnitude faster than current methods. A smaller trap capable of higher axial mode frequencies could allow for lower anomalous heating[33]. It would also allow for faster exchange transport, particularly through the low mode-frequency portion, further mitigating the effects of heating. A recent theoretical study examined exchange cooling, leveraging invariant-based engineering to develop fast transport trajectories. Those theoretical waveforms execute exchange cooling in just a few ion motional periods[24]. Combining the methodology in that work with the understanding of stray fields and electrode filtering presented here may reduce transport times significantly. Exchange cooling could be combined with existing techniques to execute entangling gates without fully merging ions[34]. Cooling of radial modes may be achieved by similar tuning of the radial mode frequencies onto resonance. For radial modes, the coupling rate in Equation (2) drops by only a factor of two, implying that relatively short cooling durations are still achievable. Alternatively, radial mode phonons could be transferred into the axial mode before axial exchange cooling[35,36]. Ref. 28 examines the exchange of phonons between chains of ions, or between a single ion and another pair, and shows that adding more ions to each chain increases the coupling rate. These scenarios could be important in QCCD architectures employing ion strings, where they could be used to cool the easily-excited axial center-of-mass mode after merge operations and prior to entangling gate operations[11]. Exchange cooling may be less effective for out-of-phase motional modes, however these modes typically suffer less from heating. Alternatively, the ions in such architectures could be cooled one by one.

## Methods
### Transport Control
To produce the electrode voltages necessary for this experiment, we employ National Instruments PXIe-5413 AWG cards. For the exchange transport, we utilize 24 of these cards to generate 48 control voltages

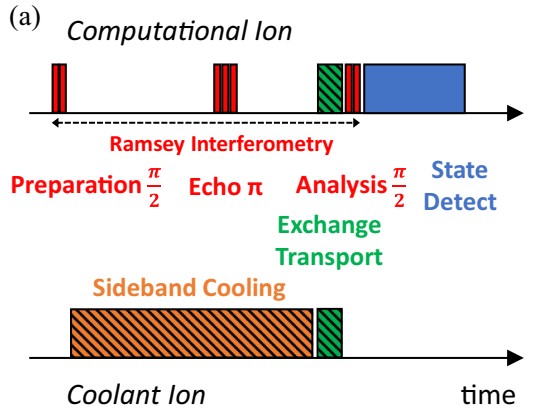

(a)

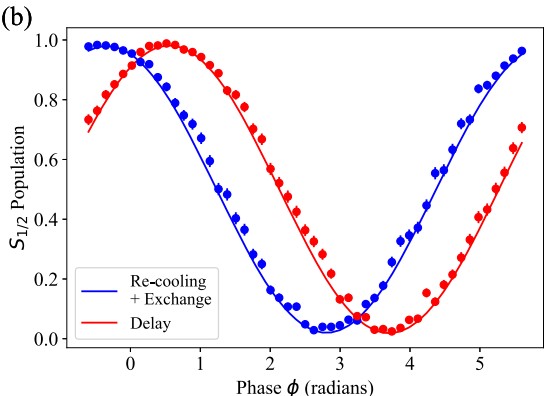

(b)

**Fig. 4 | Ramsey interferometry.** (**a**) Experimental sequence for a Ramsey experiment on the computational ion which incorporates re-cooling of the coolant ion and exchange transport (not to scale). Composite $\pi/2$ pulses for preparation and analysis begin and end the interferometry. Between $\pi/2$ pulses, sideband cooling on the coolant ion and exchange transport are performed. These two operations (marked with hash coloring) are alternatively replaced with equivalent delays. An echo composite $\pi$ pulse mitigates the impact of slow magnetic field drift. (**b**) Computational-ion Ramsey fringe, measured by varying the relative phase $\phi$ between preparation and analysis $\pi/2$ pulses. Blue points include re-cooling of the coolant ion and exchange transport. The red points include an equivalent delay. Error bars represent the 68% confidence interval in state populations assuming binomial statistics. Solid lines are fits to the data.

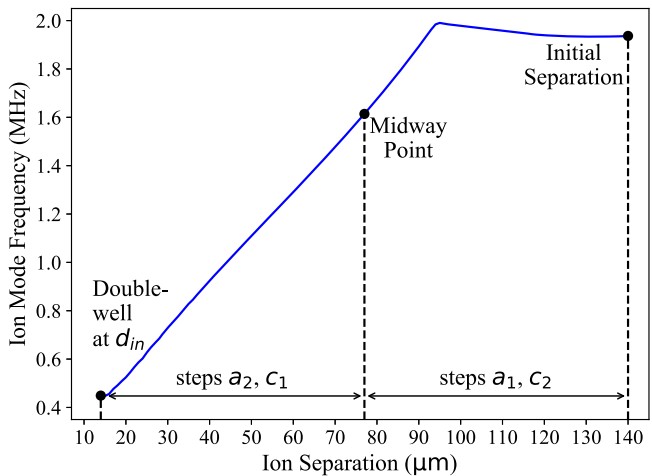

**Fig. 5 | Simulations of ion axial mode frequency.** Simulations of the computational ion mode frequency absent any Coulomb interaction as a function of double-well potential separation. Because the transport waveform is symmetric, the coolant ion mode frequency is the same.

## Table 1 | Transport Duration

| | Approach (*a*) | Duration (*µs*) |
|---|---|---|
| $a_1$ | Linear Transport | 5 |
| $a_2$ | Transport to separation ~ $d_{in}$ | 8.75 |
| $s_a$ | Settling | 17.5 |
| | **Energy Exchange (*b*)** | |
| $b_1$ | Ramp Compensating Potentials on | 2 |
| $b_2$ | Hold $t_{ex}$ | 5.8 |
| $b_3$ | Ramp Compensating Potentials off | 2 |
| $s_b$ | Settling | 7.5 |
| | **Separation (*c*)** | |
| $c_1$ | Transport from separation ~ $d_{in}$ | 31.75 |
| $c_2$ | Linear Transport | 5 |
| $s_c$ | Settling | 22 |
| | Total | 107.3 |

### Transport Sequence

Table 1 lists each step of the transport along with its associated duration. Again, the transport was optimized in three parts: *a* – approach, *b* – energy exchange and *c* – separation. The calibration procedure for part *b* is discussed in the main text. In particular, it is necessary to optimize both $E_c$ and $t_{ex}$ for step $b_2$, where the exchange time is shorter than the filter response time (Fig. 3a).

For parts *a* and *c*, we seek to transport as fast as possible while incurring minimal motional excitation. Our strategy is to break these parts down further, into steps $a_1$, $a_2$ and $c_1$, $c_2$. This partitioning enables faster transport where the ion mode frequencies remain high. Specifically, step $a_1$ transports the ions from a separation of 140 µm to 77 µm, where the ion mode frequencies remain within 20% of their maximum. The transport across this region can be performed in just 5 µs with negligible excitation. During the separation, we execute this transport in reverse for step $c_2$. At smaller ion separations, the mode frequencies become limited by the finite output range of the AWG channels and drop further[24]. We execute transport across this region more slowly, particularly in step $c_1$.

Next, we discuss our procedure for selecting settling delays $s_a$, $s_b$ and $s_c$. We determine the necessary delay in step $s_c$ by executing exchange transport followed by motional mode spectroscopy. The 22 µs delay is chosen such that the axial mode frequencies return to

on the Q0-Q31 and Q44-Q59 electrodes of the Peregrine trap[29]. Each AWG channel can generate voltages between ± 12 V and can be updated at 200 MSamples/s. The channels have 16 bit depth, or a quantization of 366 µV.

Each channel is filtered via a sixth-order Chebyshev lowpass filter designed with a cutoff frequency of 150 kHz[37]. This cutoff was chosen to provide substantial ( > 50 dB) noise suppression at the lowest mode frequencies involved in the exchange transport. Simulations of the step response of the filters show a rise to 90% of the target voltage after ~ 7.5 µs. Additional ripples settle to within 1% of the target within ~ 22 µs. These values guide the settling delays chosen after each portion of transport.

We calculate the appropriate voltages for ion transport using a boundary element method electrostatic solver and impose axial symmetry on the electrode potentials about the Q12/Q13 electrodes[38]. The solver results can be used to model potential well positions and mode frequencies, assuming static potentials for each separation, as in Fig. 5. Simulations of ion motional dynamics are discussed in Ion Motional Simulations.

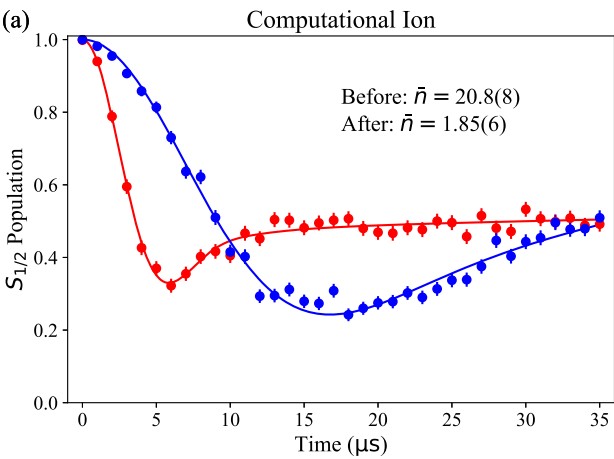

**Fig. 6 | Mode temperature characterization via sideband flopping.** Blue sideband flopping on the computational ion **a)** and coolant ion **b)** before (red) and after (blue) exchange transport. Error bars represent the 68% confidence interval in state populations assuming binomial statistics. Solid lines are fits to each flop, with the fitted $\bar{n}$ reported.

within 2 kHz of their asymptotic values. This is the mode frequency accuracy necessary for a two-qubit gate fidelity above 99.9%, assuming experimental gate parameters from Ref. 39. We begin with a similar delay for step $s_a$; however, we find that we can reduce this duration to 17.5 $\mu$s with no discernable effect on the exchange dynamics plotted in Fig. 3a (i.e. sufficient settling time to isolate parts $a$ and $b$).

For step $s_b$, the settling after the exchange, we allow 7.5 $\mu$s. This ensures enough latency for a large portion of the compensating potential ramp to fall. One consequence of this decision is an incomplete settling of the ramp off before separation (steps $s_b$ and $c_1$). At the start of separation step $c_1$, the ions are in the shallowest double-well potential and susceptible to small changes in potential and timing[6]. It is during this time that the linear and harmonic compensating potentials are still settling. This may explain the need to separate more slowly in step $c_1$ (31.75 $\mu$s) relative to step $a_2$ (8.75 $\mu$s).

### Temperature Measurements

To extract the ion mode temperatures plotted in Fig. 3b, we perform Rabi experiments in which we drive a blue motional sideband of the $S_{1/2} - D_{1/2}$ transition. We measure the $S_{1/2}$ state population $P_S$ as a function of optical pulse time $t$[40]:

$$P_S(t) = \frac{1}{2}\left(1 + \sum_{n=1}^{\infty} \frac{\bar{n}^n}{(\bar{n}+1)^{n+1}} \cos(2\Omega_{n,n+1} t)\right) \qquad (3)$$

$$\Omega_{n,n+1} = \eta \Omega_0 e^{-\eta^2/2} \sqrt{\frac{1}{n+1}} L_n^1(\eta^2) \qquad (4)$$

Here, a thermal distribution across Fock states $|n\rangle$ is assumed. $\Omega_{n,n+1}$ is the Rabi frequency for the first blue sideband transition for an ion starting in $|n\rangle$. $\Omega_{n,n+1}$ depends on the carrier optical Rabi rate $\Omega_0$, as well as $\eta$, the ion's axial mode Lamb-Dicke parameter. $L_n^1$ is the $n^{th}$ associated Laguerre polynomial of order 1. With $\Omega_0$ and $\eta$ already calibrated, we can determine $\bar{n}$ and its uncertainty by fitting the measured $S_{1/2}$ state population to Equation (3). In Fig. 6, we provide an example of these fits before and after exchange transport with a computational ion initially heated to 20.8(8) quanta.

Additionally, we corroborate the results of our sideband fits against the common sideband ratio technique[41]. The results are reported in Table 2, showing good agreement between the two methods.

For each of the measured points in Fig. 3a, we use a variant of the sideband flopping method. Rather than repeating the experiment at an

### Table 2 | Exchange cooling performance

| Initial $\bar{n}$ (quanta) | Final $\bar{n}$ (quanta) | Final $\bar{n}$ SBR (quanta) |
|---|---|---|
| 0.51 (3) | 1.69 (6) | 2.0 (4) |
| 15.0 (6) | 2.15 (8) | 1.9 (4) |
| 20.8 (8) | 1.85 (6) | 3.0 (8) |
| 26 (1) | 2.01 (8) | 2.2 (5) |
| 37 (2) | 2.6 (2) | 2.3 (6) |
| 44 (2) | 2.5 (1) | 3.3 (10) |
| 53 (3) | 2.6 (1) | 2.0 (5) |
| 68 (3) | 3.0 (1) | 4 (1) |
| 78 (4) | 3.5 (2) | 3.3 (10) |
| 95 (4) | 3.5 (2) | 3.7 (12) |
| 106 (5) | 3.9 (2) | 4 (1) |

Computational ion temperature before and after exchange transport. Initial temperature is measured using the sideband flopping technique. The temperature after exchange is measured using both the flopping technique and the sideband ratio (SBR) technique.

assortment of blue sideband pulse durations, we use a fixed $\tau = 5$ $\mu$s pulse. This generates a value of $P_S(\tau)$ for each value of $t_{ex}$. To calculate the ion temperature, we numerically solve for $\bar{n}$ by inverting Equation (3), again having already calibrated the other parameters in the equation. Error bars are calculated by solving for $\bar{n}$ at $P_S(\tau) \pm \sigma$, where $\sigma$ is the error in $P_S(\tau)$.

### Double-well Modeling

For an effective energy exchange, we require fine manipulation of the potential through part $b$. Hence, we seek to understand the discepancies between the designed resonant potential at target ion separation $d_{in}$, and the potential actually generated in the trap. Considerable deviations from design are expected due to stray fields, fabrication imperfections, AWG inaccuracies, and inexact electrostatic trap models. For example, such deviations have previously been observed and quantified in the context of an ion merge operation[6]. For this exercise, we use a simplified model of the trap confinement, considering motion only along the trap axis $z$. We begin with a fourth-order polynomial model describing an arbitrary axial potential[42,43]:

$$V(z) = -E_0 z + \alpha z^2 + \gamma z^3 + \beta z^4. \qquad (5)$$

An ideal 1D double-well potential can be expressed using just harmonic ($\alpha$), and quartic ($\beta$) terms. Additional terms $E_0$ and $\gamma$ represent

undesired modifications from linear and cubic components, respectively.

In design, we calculate voltages from the boundary element solver to hold the ions resonantly at separation $d_{in}$. We fit the potential along an axial path, with fixed height $y$ and lateral dimension $x$, to Equation (5). This yields coefficients $\alpha^* = -0.3$ V/mm$^2$ and $\beta^* = 9 \times 10^3$ V/mm$^4$. We call this 1D approximation $V(z)$. Because of the imposed symmetry about the Q12/Q13 electrodes, the linear and cubic terms are negligible. For our experiment, we approximate the potential as one dimensional and call it $V'(z)$. To characterize $V'(z)$, we use a single measurement of $\Omega_+$, $\Omega_-$ and the ion axial positions $z_1$ and $z_2$ at $E_c = 10.5$ V/m and $\alpha_c = 0$ V/mm$^2$ (see Fig. 2). The imaging magnification and $z = 0$ pixel position are determined from a reference image taken of two ions separated by 140 μm. With $\Omega_+$ and $\Omega_-$, $z_1$ and $z_2$, the coefficients of the potential $V'(z)$ are fully defined: $\alpha' = -1.3$ V/mm$^2$, $\beta' = 8.4 \times 10^3$ V/mm$^4$, $\gamma' = -1.2 \times 10^2$ V/mm$^3$, $E'_0 = -23$ V/m. Here, we have assumed that the added compensating potential is locally ideal.

Starting from $V'(z)$, it is necessary to apply both a linear $(V(z) = E'_0 z)$ and cubic compensating potential $(V(z) = -\gamma' z^3)$ to recover a potential which is symmetric about $z = 0$, such as $V'(z)$. However, with arbitrary control over the linear potential as outlined above $(V(z) = E_c z)$, it is possible to generate a double-well potential which achieves resonance without cubic compensation. A symmetric double-well potential, centered about axial position $z_0 = -\gamma'/(4\beta')$, can be achieved with $E_c = -E'_0 + 2z_0(\alpha' + \alpha_c + z_0\gamma')$. The linear relationship between $\alpha_c$ and resonant $E_c$ is evident in the diagonal blue stripe of Fig. 2b.

## Ion Motional Simulations

We utilize a classical simulator to model our ion motional dynamics through the exchange transport. The simulator employs a 3D electrostatic model of the ion trap. To calculate ion dynamics, we provide the simulator with voltage waveforms for each of the quasi-static electrodes. We start with the waveforms as in experiment: a sum of the designed waveform from the boundary element solver and any applied compensating potentials. Then in simulation, we add a correction waveform. The correction is meant to capture experimental deviations in the resonant exchange potential calibrated as in Double-well Modeling. The aggregate waveforms are digitally filtered to model the effect of our lowpass filters. The simulator solves for each ion's position and velocity using an energy-conserving symplectic integrator. RF confining fields are simulated explicitly rather than via the pseudopotential approximation. The simulations do not model ion heating.

Here, we discuss the procedure for generating the correction waveform. In a one dimensional approximation, this waveform should correct axial potential $V(z)$ to match $V'(z)$:

$$V'(z) = \underbrace{-E'_0 z + (\alpha' - \alpha^*)z^2 + \gamma' z^3 + (\beta' - \beta^*)z^4}_{\text{Correction terms}} + V^*(z) \tag{6}$$

To make the corrections for simulation, we use our boundary-element solver to calculate the voltages necessary for linear, quadratic, cubic and quartic axial correction potentials. Correction potentials apply zero field to each ion in the radial directions. The corresponding correction voltages are added to the waveforms for simulation of step $s_a$ and part $b$. Because these correction potentials form only a local approximation, the correction voltages are ramped on (off) linearly in step $a_2$ ($c_1$) starting (ending) when the designed ion separation reaches $2d_{in}$.

To generate theoretical curves in Fig. 3a, we simulate all of the steps in Table 1 and calculate each ion's final classical energy. The coolant ion is always initialized with zero velocity. To model the initial thermal state of the computational ion, we simulate the trajectory many times for each value of $t_{ex}$, choosing a different energy for the computational ion each time. With the simulated runs, we perform a

weighted average of the final classical energy of each ion. The weights correspond to a Boltzmann distribution of initial computational ion energy with temperature $\hbar\omega_1\bar{n}/k_b$. Here, $k_b$ is Boltzmann's constant and $\bar{n} = 15$ quanta. We divide the weighted final classical energy by $\hbar\omega_{1,2}$ to make a comparison with the measured mean energy $\bar{n}$.

As stated in the main text, we notice a day-to-day drift of a few V/m in the optimal value of resonant $E_c$ used in step $b_2$. To choose a single resonant $E_c$ for simulation, we follow a similar procedure to experiment. We select a value of $E_c$ which optimizes the first simulated exchange in Fig. 3a. This allows us to account for any drift in $E_c$ between acquiring the correction calibration data in Fig. 2 and the exchange data in Fig. 3. On the days during which the data in Fig. 3 was taken, there is a discrepancy of ~ 3 V/m between optimized experiment and simulation. This is reasonable given the observed range of experimental drift.

Following this line of thinking, we diagnose the imperfect energy transfer observed in Fig. 3b. We suspect the ~ 2% fraction of energy which is not transferred to the coolant ion is likely the result of imprecision in our ability to set $E_c$ through the exchange. In simulation, an inaccuracy of 0.7 V/m causes a large enough deviation from resonance to reduce $A_{ex}$ by 2%. This is due to the anharmonicity of the double-well potential, which leads to an increase in one ion's frequency and a decrease in the other's when they experience an identical (static) force displacing them from equilibrium. In experiment, the quantization of our electrode voltage sources allows for at best 1 V/m precision in $E_c$ (see Transport Control).

To generate Fig. 2c, d, we simulate transport steps $a_1$ through $b_2$ from Table 1. For each combination of $E_c$ and $\alpha_c$ we record the simulated ion separation in Fig. 2d. To determine $\Omega_+$ and $\Omega_-$, we perform a Fourier transform of the ions' axial velocities. We extract the two strongest frequency components and plot their difference $\delta\Omega$ in Fig. 2c.

In a QCCD architecture, after loading and initial laser cooling of relevant computational ion modes nearly to the ground state, it is important to keep their temperatures within the Lamb-Dicke regime during the course of a calculation. However, in infrequent circumstances, such as when traversing between ion trap chips[32] or when attempting ion merge/separation with a lost ion, the computational ion temperature may increase dramatically. We have used our classical simulator to study the effectiveness of exchange cooling for axial mode temperatures well outside the Lamb-Dicke regime. In these situations, the computational ion's motion samples a larger portion of the electric potential where anharmonicities could potentially diminish its effectiveness. Provided that one has sufficient control over $E_c$, our simulations indicate that the technique is still capable of sub-quanta cooling with initial temperatures as high as $\bar{n} \approx 1000$ quanta. It is of interest to note that, in contrast with traditional sympathetic cooling, the cooling duration is independent of initial temperature.

## Ramsey Sequence

To investigate the effect of re-cooling on the internal coherence of the computational ion, we utilize Ramsey interferometry as shown in Fig. 4a and Fig. 7. Our beam geometry, in particular the use of a global 854 nm laser beam, dictates that we store the computational qubit in the $|-1/2\rangle = S_{1/2}(m_J = -1/2)$ and $|+1/2\rangle = S_{1/2}(m_J = +1/2)$ Zeeman manifold. In this way, we can execute sideband cooling, including deshelving pulses on the coolant ion, without disturbing the computational ion. Separate 854 nm laser beams would enable this technique with the $S_{1/2} - D_{5/2}$ optical qubit.

We prepare our test by Doppler cooling both ions and then sideband cooling the computational ion while maintaining 140 μm ion separation. Next, we prepare both ions in the $|-1/2\rangle$ state. To examine Zeeman qubit coherence, we make use of the state $|D\rangle = D_{5/2}(m_J = -1/2)$ in the protocol in Box 1.

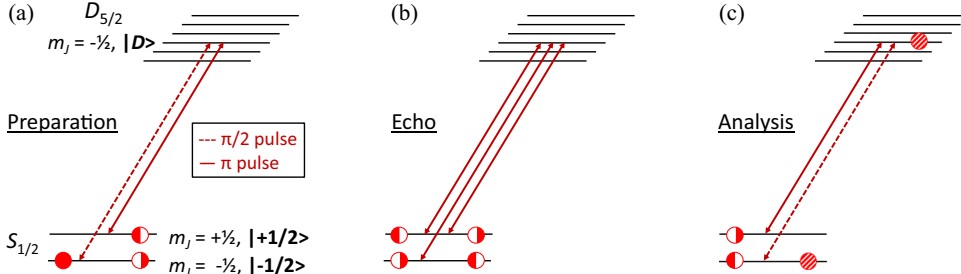

**Fig. 7 | Population transfer scheme for ramsey interferometry.** 729 nm laser pulses on the computational ion used for Ramsey interferometry, executed from left to right. The experiment is comprised of three sets of pulses, (**a**) preparation, (**b**) echo and (**c**) analysis, described in Box 1. Circles indicate the ion state population before and after each set of pulses. The filled circle represents complete population in the $|-1/2\rangle$ state. Half circles indicate 50/50 superpositions between the $|+1/2\rangle$, $|-1/2\rangle$ states. Population swaps between these two states during the echo sequence. Hashed circles represent a superposition of the $|-1/2\rangle$ and $|D\rangle$ states.

## BOX 1
# Ramsey Sequence

1. Preparation, Composite $\pi/2$.
   Two pulses on the computational ion create an equal superposition of the $|-1/2\rangle$ and $|+1/2\rangle$ states: a $\pi/2$ pulse on the $|-1/2\rangle \leftrightarrow |D\rangle$ transition, followed by a $\pi$ pulse on the $|D\rangle \leftrightarrow |+1/2\rangle$ transition.
2. Re-cooling and Echo
   (a) Sideband cooling on the coolant ion. We employ the 'varied-width' approach to sideband cooling, alternating red sideband $\pi$ pulses with 854 and 866 pulses, beginning at target Fock state $|n = 45\rangle$ [44].
   (b) Echo, Composite $\pi$. A sequence of three pulses on the computational ion swap population in the $|-1/2\rangle$ and $|+1/2\rangle$ states: consecutive $\pi$ pulses on $|+1/2\rangle \leftrightarrow |D\rangle$, $|-1/2\rangle \leftrightarrow |D\rangle$, and $|+1/2\rangle \leftrightarrow |D\rangle$ transitions. The pulse sequence is executed at time $(t_{SBC} + t_{RT})/2$ after the preparation pulses ($t_{SBC}$ and $t_{RT}$ are the durations of sideband cooling and round-trip exchange transport, respectively).
3. Exchange Transport
4. Analysis, Composite $\pi/2$.
   A reversal of the initial pulses on the computational ion return populations to the $|-1/2\rangle$, $|D\rangle$ basis: a $\pi$ pulse on the $|+1/2\rangle \leftrightarrow |D\rangle$ transition followed by a $\pi/2$ pulse on the $|D\rangle \leftrightarrow |-1/2\rangle$ transition.
5. State Detection

We repeat this process, varying the phase $\phi$ between the first $\pi/2$ pulse and last $\pi/2$ pulse, to create the Ramsey fringes in Fig. 4b. In the case of the red curve, we replace steps 2a and 3 with equivalent delays. We fit each curve to a sine wave, $A\sin(\phi + \phi_0) + \frac{1}{2}$, and extract the amplitude $A$. Phase $\phi_0$ is an overall offset. The contrast is limited by the quality of these pulses and by our ion coherence after ~1 ms in the unshielded magnetic environment.

## Data availability
The data presented in this manuscript are available from the corresponding author upon request.

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

## Acknowledgements

The authors thank Jonathan Andreasen and J. True Merrill for development of and assistance with the ion motional simulations. We thank Christopher M. Shappert for help with trap installation and cryogenic system operation. This work was done in collaboration with Los Alamos National Laboratory.

## Author contributions

S.D.F, V.S.S, R.A.M. and H.N.T. assembled the experiment apparatus. The data was collected by S.D.F, V.S.S and J.M.G. Simulations and analyses were performed by S.D.F, J.M.G, C.R.C. and K.R.B. S.D.F. wrote the manuscript with input from all authors. The project was led by C.R.C. and K.R.B.

## Competing interests

The authors declare no competing interests.
