## [Peer Review File · Nature Communications]

Rapid Exchange Cooling with Trapped IonsREVIEWER COMMENTS

Reviewer #1 (Remarks to the Author):

This manuscript describes experiments carried out in the context of scalable methods for trapped-ion quantum computing. Present implementations of multi-qubit gates explore the joint motion of the qubit ions under the influence of their Coulomb interaction and of the external trapping potential to mediate the required effective interaction between the qubits. To varying degrees, the performance of two-qubit gates depends on the motional-state occupation of the state of motion of the ions prior to the gate. Sympathetic cooling of the qubit ion with a different ion species (cooling ion) is typically considered to re-initialize the motion close to the ground state prior to any two-qubit gate. A common issue in implementing sympathetic cooling schemes is the potentially challenging cooling dynamics for qubit / cooling ion combinations of different masses and the time needed for sympathetic cooling. Both effects are at least partially related.

This manuscript describes a transport-based method where the cooling ion can be of the same species as the qubit ion. The laser cooling happens when the cooling ion is well separated from the qubit ion. The motional energy of the qubit ion is then reduced by swapping the motional states of the cooling ion and the qubit ion after they have been brought into a double-well potential.

This is a very interesting and novel result. I enjoyed reading the manuscript. It addresses a key challenge in scaling ion-trap quantum computers based on the QCCD architecture. Though the challenges of implementing sympathetic cooling in any given scenario for the QCCD architecture are very much a function of the type of gate and motional mode employed, of its sensitivity to the motional-state occupation and of the particular choice of species, sympathetic cooling is a task of considerable importance in scaling. The method presented in this manuscript is not free of its own challenges because it requires very careful potential engineering, but it is definitely worth pursuing. The demonstrated final-state population may still have to be improved depending on the scenario, but the effect and method is clearly demonstrated. Issues to be considered before these results can be applied in a computational context e.g. for a two-qubit gate would be how to cool a pair of qubit ions and how to cool the multiple motional modes, in particular the radial ones. The authors could possibly include a little more in-depth discussion of these questions and how this cooling scheme might play out in the context of a multi-register QCCD quantum processor to stimulate the discussion about this question in the literature.

The material supports the conclusions and claims; the analysis is very clear. The methods used to extract the data are well accepted in the field.

I fully recommend publication of the manuscript.

Reviewer #2 (Remarks to the Author):

In their manuscript “Rapid Exchange Cooling with Trapped Ions,” Fallek et al. describe a set of experiments where energy from a computational trapped-ion qubit is quickly offloaded into a bank of continuously laser-cooled ions. Compared to standard sympathetic cooling, this approach is demonstrated to be approximately an order-of-magnitude faster while removing most of the axial energy from the computational ion. Furthermore, the authors verify that their exchange cooling process does not introduce decoherence on the computational ion.

Cooling during a trapped-ion quantum computation has long been identified as a critical process within a quantum charge-coupled device (QCCD) architecture. This need for fast and efficient cooling arises from imperfect QCCD transport protocols during a computation, as well as other noise sources, which lead to unwanted ion motion mid-circuit. Historically, sympathetic cooling with co-trapped ions of a different species have been used to address this heating, at the cost of slower circuit runtimes and increased experimental complexity.

The authors’ development of “rapid exchange cooling,” while still a type of sympathetic cooling, addresses both the slow runtime and multi-species complexity issues of historical sympathetic cooling. Their procedure involves carefully shuttling the computational ion into the proximity of a coolant ion/ions so that motional quanta may be coherently exchanged, then returning the computational ion to its original position.

The protocol is very well described in the text. The data supports the premise that hot computational ions may be cooled back near the ground state much more quickly than traditional sympathetic cooling, and that the recooling process does not introduce decoherence effects on the computational ion.

My primary concern with this manuscript is its applicability to state-of-the-art QCCD devices. Already, transport waveforms within ion traps can limit heating to just a few motional quanta. Recent work (PRL 130, 173202 (2023)) has also demonstrated transport through QCCD junctions with only 0.01-0.03 quanta of excitation. Thus, this manuscript’s demonstration of cooling from 10’s-100+ quanta down to the few-quanta level targets quite a different regime than the heating which arises from shuttling operations or from fluctuating electric fields. The inability to cool below the few-quanta level (shown in Fig. 3b and discussed in the surrounding text), due to hardware limitations in precisely controlling the trapping fields, may ultimately limit this technique’s utility in reducing the already small levels of heating experienced in leading-edge QCCD architectures.

Reviewer #3 (Remarks to the Author):

The article describes how the principle of coherent energy transfer between coupled oscillators can be used in the context of trapped-ion quantum technology to cool qubit ions below the Doppler cooling temperature by using auxiliary ground-state cooled ions. The article carefully describes a full cooling protocol for two ions comprising three fundamental steps. First how to bring together a "hot" qubit ion with a "cold" coolant ion. Second, how to tune their interaction time to obtain optimal cooling of the qubit ion. Third, how to separate them after cooling. The description of the experimental procedures is clear and well-supported by the data. The data shows a 10-fold improvement in the cooling speed as compared to other state-of-the-art methods for cooling ions near the motional ground state such as sideband cooling. This is a substantial speed-up that makes the technique demonstrated in this article appealing in the context of trapped-ion quantum devices. I found the result innovative and worth publication in Nature Communication.

Although I believe that the results are interesting and valid, I have two reservations.

1) The authors present and discuss experimental results for two ions. Several QCCD architectures foresee trapping sites that include several ions co-trapped together and the authors should suggest a roadmap to apply the presented technique to a chain composed of several ions. A few questions that may help the authors formulating better their point of view. Should the ions be individually cooled one by one? Can several qubit ions be cooled simultaneously via a single one? How would the requirement of cooling several ions affect the cooling time? Would this technique still be faster than other techniques such as EIT cooling that can address several motional modes simultaneously? What are the authors' recommendations for scaling?

2) In the presented measurements and experimental conditions, the authors do not mention nor calculate the maximum starting temperature that the "hot" ion can have for the method to be utilized. I think this is a crucial point that should be directly and quantitatively addressed because it is fundamental for the application of the technique in a quantum device.

2.1) pag.4: " For small oscillations of the ions about their equilibrium positions, A_{ex} is given by:". How small? Can this "small" be compared to other quantities? Does the presented formula rely on some Taylor expansion?

2.2) As a reader, I cannot understand if the comparison with sympathetic cooling is fully correct. The comparison starts on page 2 with "In this work, we explore exchange cooling, an alternative to sympathetic cooling". Sympathetic cooling of ions is a general technique that applies to ions at any initial temperature, not just when they are already crystallized or perform small oscillations around their equilibrium positions. Is the maximum temperature a relevant point in which sympathetic cooling and the presented exchange cooling technique differ? Does this imply a different applicability area? Is it

meaningful to discuss in the abstract and in the introduction that the technique is relevant only for already crystalized ions?

2.3) Following on 2.2. The technique appears relevant for quantum devices for cooling ions that do not possess favorable electronic transitions on which standard laser cooling cannot be applied. In fact, the authors present formulas for ions with different masses on page 3. Is the technique relevant for "hot" ions cannot that cannot be Doppler laser cooled? Does the technique require other pre-cooling methods in case the qubits are too "hot"? If so, at which temperature are they too "hot"?

2.4) I found a statement of the article to be incomplete because the maximum starting temperature is not stated: pag. 5 "In the hottest case, we reduce the computational ion temperature by 102(5)...". In this context, the authors should write in the sentence the number of how many phonon excitations correspond to the condition to be the "hottest". Besides, I was wondering, is this "hottest" the "hottest" achievable because at higher temperatures the technique does not work any longer?

To conclude, I think that addressing my comments will not change the presented result nor it will affect their validity and impact. I recommend the article for publication in Nature Communication after revision.

Spencer Fallek
Response to Reviewers for 'Rapid Exchange Cooling with Trapped Ions'
Manuscript NCOMMS-23-44961-T

Dear Editor,

Below we respond point-by-point to the comments of the referees. We are also providing two copies of the manuscript, one clean version and one with all modifications highlighted.

Reviewer #1 (Remarks to the Author):

This manuscript describes experiments carried out in the context of scalable methods for trapped-ion quantum computing. Present implementations of multi-qubit gates explore the joint motion of the qubit ions under the influence of their Coulomb interaction and of the external trapping potential to mediate the required effective interaction between the qubits. To varying degrees, the performance of two-qubit gates depends on the motional-state occupation of the state of motion of the ions prior to the gate. Sympathetic cooling of the qubit ion with a different ion species (cooling ion) is typically considered to re-initialize the motion close to the ground state prior to any two-qubit gate. A common issue in implementing sympathetic cooling schemes is the potentially challenging cooling dynamics for qubit / cooling ion combinations of different masses and the time needed for sympathetic cooling. Both effects are at least partially related.

This manuscript describes a transport-based method where the cooling ion can be of the same species as the qubit ion. The laser cooling happens when the cooling ion is well separated from the qubit ion. The motional energy of the qubit ion is then reduced by swapping the motional states of the cooling ion and the qubit ion after they have been brought into a double-well potential.

This is a very interesting and novel result. I enjoyed reading the manuscript. It addresses a key challenge in scaling ion-trap quantum computers based on the QCCD architecture. Though the challenges of implementing sympathetic cooling in any given scenario for the QCCD architecture are very much a function of the type of gate and motional mode employed, of its sensitivity to the motional-state occupation and of the particular choice of species, sympathetic cooling is a task of considerable importance in scaling. The method presented in this manuscript is not free of its own challenges because it requires very careful potential engineering, but it is definitely worth pursuing. The demonstrated final-state population may still have to be improved depending on the scenario, but the effect and method is clearly demonstrated. Issues to be considered before these results can be applied in a computational context e.g. for a two-qubit gate would be how to cool a pair of qubit ions and how to cool the multiple motional modes, in particular the radial ones. The authors could possibly include a little more in-depth discussion of these questions and how this cooling scheme might play out in the context of a multi-register QCCD quantum processor to stimulate the discussion about this question in the literature.

The material supports the conclusions and claims; the analysis is very clear. The methods used to extract the data are well accepted in the field.

I fully recommend publication of the manuscript.

We thank the referee for his/her careful review. Here are our responses to the comments:

Suggestion: "Issues to be considered before these results can be applied in a computational context e.g. for a two-qubit gate would be how to cool a pair of qubit ions and how to cool the multiple motional modes, in particular the radial ones..."

We have modified the Discussion section with the highlighted edits below. These changes further discuss radial mode cooling as well as cooling pairs or longer chains of ions.

“Cooling of radial modes may be achieved by similar tuning of the radial mode frequencies onto resonance. For radial modes, the coupling rate in Equation 2 drops by only a factor of two, implying that relatively short cooling durations are still achievable. Alternatively, radial mode phonons could be transferred into the axial mode before axial exchange cooling [34, 35]. Ref. [27] examines the exchange of phonons between chains of ions, or between a single ion and another pair, and shows that adding more ions to each chain increases the coupling rate. These scenarios could be important in QCCD architectures employing ion strings, where they could be used to cool the easily-excited axial center-of-mass mode after merge operations and prior to entangling gate operations [11]. Exchange cooling may be less effective for out-of-phase motional modes, however these modes typically suffer less from heating. Alternatively, the ions in such architectures could be cooled one by one.”

The newly added Ref. 35 discusses a successful implementation of swapping phonons from a radial mode into an axial mode prior to laser cooling.

[35] Hou, P.-Y. et al. Indirect Cooling of Weakly Coupled Trapped-Ion Mechanical Oscillators (2023). ArXiv:2308.05158 [physics, physics:quant-ph].

Reviewer #2 (Remarks to the Author):

In their manuscript "Rapid Exchange Cooling with Trapped Ions," Fallek et al. describe a set of experiments where energy from a computational trapped-ion qubit is quickly offloaded into a bank of continuously laser-cooled ions. Compared to standard sympathetic cooling, this approach is demonstrated to be approximately an order-of-magnitude faster while removing most of the axial energy from the computational ion. Furthermore, the authors verify that their exchange cooling process does not introduce decoherence on the computational ion.

Cooling during a trapped-ion quantum computation has long been identified as a critical process within a quantum charge-coupled device (QCCD) architecture. This need for fast and efficient cooling arises from imperfect QCCD transport protocols during a computation, as well as other noise sources, which lead to unwanted ion motion mid-circuit. Historically, sympathetic cooling with co-trapped ions of a different species have been used to address this heating, at the cost of slower circuit runtimes and increased experimental complexity.

The authors' development of "rapid exchange cooling," while still a type of sympathetic cooling, addresses both the slow runtime and multi-species complexity issues of historical sympathetic cooling. Their procedure involves carefully shuttling the computational ion into the proximity of a coolant ion/ions so that motional quanta may be coherently exchanged, then returning the computational ion to its original position.

The protocol is very well described in the text. The data supports the premise that hot computational ions may be cooled back near the ground state much more quickly than traditional sympathetic cooling, and that the recooling process does not introduce decoherence effects on the computational ion.

My primary concern with this manuscript is its applicability to state-of-the-art QCCD devices. Already, transport waveforms within ion traps can limit heating to just a few motional quanta. Recent work (PRL 130, 173202 (2023)) has also demonstrated transport through QCCD junctions with only 0.01-0.03 quanta of excitation. Thus, this manuscript's demonstration of cooling from 10's-100+ quanta down to the few-quanta level targets quite a different regime than the heating which arises from shuttling operations or from fluctuating electric fields. The inability to cool below the few-quanta level (shown in Fig. 3b and discussed in the surrounding text), due to hardware limitations in precisely controlling the trapping fields, may ultimately limit this technique's utility in reducing the already small levels of heating experienced in leading-edge QCCD architectures.

We thank the referee for his/her careful review. Here are our responses to the comments:

Suggestion: "My primary concern with this manuscript is its applicability to state-of-the-art QCCD devices. Already, transport waveforms within ion traps can limit heating to just a few motional quanta..."

Indeed, the referenced publication [A] shows junction transport with low heating, albeit at speeds far below that of linear transport [B]. However, the junction work alone does not capture the levels of heating present in state-of-the-art QCCD systems, such as the Race Track H2 System [C] from the same group.

In that work [C], the coolant ions are Doppler cooled continuously during all transport operations. Temperatures of tens of quanta, near the Doppler limit, are to be expected after each transport. In similar systems, ion heating of hundreds of quanta per second is reported [D], whereas algorithms currently take as long as 20.6 seconds (see Table 1 of [C]). In a similar vein, we are not aware of any demonstrations of ion merge and separation operations with heating below the few quanta level [E]. Hence, temperatures into the tens of quanta regime should certainly be expected in modern devices.

- A. Burton, W. C. et al. Transport of Multispecies Ion Crystals through a Junction in a Radio-Frequency Paul Trap. *Physical Review Letters* 130, 173202 (2023).
- B. Sterk, J. D. et al. Closed-loop optimization of fast trapped-ion shuttling with sub-quanta excitation. *npj Quantum Information* 8, 68 (2022).
- C. Moses, S. A. et al. A Race Track Trapped-Ion Quantum Processor (2023). ArXiv:2305.03828 [quant-ph]
- D. Pino, J. M. et al. Demonstration of the trapped-ion quantum CCD computer architecture. *Nature* 592, 209–213 (2021)
- E. Ruster, T. et al. Experimental realization of fast ion separation in segmented Paul traps. *Physical Review A* 90, 033410 (2014).

Reviewer #3 (Remarks to the Author):

The article describes how the principle of coherent energy transfer between coupled oscillators can be used in the context of trapped-ion quantum technology to cool qubit ions below the Doppler cooling temperature by using auxiliary ground-state cooled ions. The article carefully describes a full cooling protocol for two ions comprising three fundamental steps. First how to bring together a "hot" qubit ion with a "cold" coolant ion. Second, how to tune their interaction time to obtain optimal cooling of the qubit ion. Third, how to separate them after cooling. The description of the experimental procedures is clear and well-supported by the data. The data shows a 10-fold improvement in the cooling speed as compared to other state-of-the-art methods for cooling ions near the motional ground state such as sideband cooling. This is a substantial speed-up that makes the technique demonstrated in this article appealing in the context of trapped-ion quantum devices. I found the result innovative and worth publication in Nature Communication.

Although I believe that the results are interesting and valid, I have two reservations.

1) The authors present and discuss experimental results for two ions. Several QCCD architectures foresee trapping sites that include several ions co-trapped together and the authors should suggest a roadmap to apply the presented technique to a chain composed of several ions. A few questions that may help the authors formulating better their point of view. Should the ions be individually cooled one by one? Can several qubit ions be cooled simultaneously via a single one? How would the requirement of cooling several ions affect the cooling time? Would this technique still be faster than other techniques such as EIT cooling that can address several motional modes simultaneously? What are the authors' recommendations for scaling?

2) In the presented measurements and experimental conditions, the authors do not mention nor calculate the maximum starting temperature that the "hot" ion can have for the method to be utilized. I think this is a crucial point that should be directly and quantitatively addressed because it is fundamental for the application of the technique in a quantum device.

2.1) pag.4: " For small oscillations of the ions about their equilibrium positions, A_{ex} is given by:". How small? Can this "small" be compared to other quantities? Does the presented formula rely on some Taylor expansion?

2.2) As a reader, I cannot understand if the comparison with sympathetic cooling is fully correct. The comparison starts on page 2 with "In this work, we explore exchange cooling, an alternative to sympathetic cooling". Sympathetic cooling of ions is a general technique that applies to ions at any initial temperature, not just when they are already crystallized or perform small oscillations around their equilibrium positions. Is the maximum temperature a relevant point in which sympathetic cooling and the presented exchange cooling technique differ? Does this imply a different applicability area? Is it meaningful to discuss in the abstract and in the introduction that the technique is relevant only for already crystallized ions?

2.3) Following on 2.2. The technique appears relevant for quantum devices for cooling ions that do not possess favorable electronic transitions on which standard laser cooling cannot be applied. In fact, the authors present formulas for ions with different masses on page 3. Is the technique relevant for "hot" ions cannot that cannot be Doppler laser cooled? Does the technique require other pre-cooling methods in case the qubits are too "hot"? If so, at which temperature are they too "hot"?

2.4) I found a statement of the article to be incomplete because the maximum starting temperature is not stated: pag. 5 "In the hottest case, we reduce the computational ion temperature by $10^2(5)$...". In this context, the authors should write in the sentence the number of how many phonon excitations correspond to the condition to be the "hottest". Besides, I was wondering, is this "hottest" the "hottest" achievable because at higher temperatures the technique does not work any longer?

To conclude, I think that addressing my comments will not change the presented result nor it will

affect their validity and impact. I recommend the article for publication in Nature Communication after revision.

We thank the referee for his/her careful review. Here are our responses to the comments:

Suggestion 1: "Several QCCD architectures foresee trapping sites that include several ions co-trapped together and the authors should suggest a roadmap to apply the presented technique to a chain composed of several ions..."

To address the comments, we added the following sentences to the Discussion section. These changes discuss cooling of chains of ions, whether a single ion can be used, and expected duration.

"Ref. [27] examines the exchange of phonons between chains of ions, or between a single ion and another pair, and shows that adding more ions to each chain increases the coupling rate. These scenarios could be important in QCCD architectures employing ion strings, where they could be used to cool the easily-excited axial center-of-mass mode after merge operations and prior to entangling gate operations [11]. Exchange cooling may be less effective for out-of-phase motional modes, however these modes typically suffer less from heating. Alternatively, the ions in such architectures could be cooled one by one."

We look forward to studying the theoretical and scaling implications of this technique with longer ion chains, including a detailed comparison of exchange cooling with other cooling schemes. However, such a detailed study would be heavily dependent on the chosen QCCD architecture and is beyond the scope of the present work.

Suggestion 2: "The authors do not mention nor calculate the maximum starting temperature that the "hot" ion can have for the method to be utilized."

We have included a paragraph in Methods Section 3.5 where we investigate the limits of this protocol as the computational ion gets hotter:

"In a QCCD architecture, after loading and initial laser cooling of relevant computational ion modes nearly to the ground state, it is important to keep their temperatures within the Lamb-Dicke regime during the course of a calculation. However, in infrequent circumstances, such as when traversing between ion trap chips [31] or when attempting ion merge/separation with a lost ion, the computational ion temperature may increase dramatically. We have used our classical simulator to study the effectiveness of exchange cooling for axial mode temperatures well outside the Lamb-Dicke regime. In these situations, the computational ion's motion samples a larger portion of the electric potential where anharmonicities could potentially diminish its effectiveness. Provided that one has sufficient control over E_c , our simulations indicate that the technique is still capable of sub-quanta cooling with initial temperatures as high as $\bar{n} \approx 1000$ quanta. It is of interest to note that, in contrast with traditional sympathetic cooling, the cooling duration is independent of initial temperature."

Suggestion 2.1. "For small oscillations of the ions about their equilibrium positions, A_{ex} is given by:'. How small? Can this "small" be compared to other quantities? Does the presented formula rely on some Taylor expansion?"

Yes, it does depend on a Taylor expansion. This approximation is based upon Equations 1 and 2 of Reference 25: *Brown, K. R. et al. Coupled quantized mechanical oscillators. Nature 471, 196–199 (2011)*. We modified that sentence and included a citation: "For small oscillations of the ions about their equilibrium positions, when the potential can be approximated to second order in the ion displacements [25], A_{ex} is given by..."

Suggestion 2.2. "I cannot understand if the comparison with sympathetic cooling is fully correct. ... Is the maximum temperature a relevant point in which sympathetic cooling and the presented exchange cooling technique differ? Does this imply a different applicability area? Is

it meaningful to discuss in the abstract and in the introduction that the technique is relevant only for already crystalized ions?"

In QCCD architectures, it is often assumed that the computational ions are initially trapped and cooled via direct Doppler cooling. This is followed by direct sideband cooling or sideband cooling of the sympathetic ions to reduce the relevant mode temperatures from tens of quanta to near the ground state. We highlight this in the added sentence above: "In a QCCD architecture, after loading and initial laser cooling of relevant computational ion modes nearly to the ground state, it is important to keep their temperatures within the Lamb-Dicke regime during the course of a calculation."

Additionally, the references below (a-c) discuss this scheme. Hence, we do not feel that in this context, it is necessary to highlight the differences for un-crystallized ions. For QCCD systems the applicability area should be the same for both cooling methods (tens of quanta and below).

- a. Pino, J. M. et al. Demonstration of the trapped-ion quantum CCD computer architecture. *Nature* 592, 209–213 (2021)
- b. Wan, Y. et al. Quantum gate teleportation between separated qubits in a trapped-ion processor. *Science* 364, 875–878 (2019).
- c. Hilder, J. et al. Fault-Tolerant Parity Readout on a Shuttling-Based Trapped-Ion Quantum Computer. *Physical Review X* 12, 011032 (2022).

Suggestion 2.3. "Is the technique relevant for "hot" ions cannot that cannot be Doppler laser cooled? Does the technique require other pre-cooling methods in case the qubits are too "hot"? If so, at which temperature are they too "hot"?"

In the scenario where the computational ion cannot be directly laser cooled, this technique may still be applicable. As discussed in the added paragraph above, there are limits to how hot the computational ion can be, but these appear rather high. Indeed, it is possible to use this technique with coolant ions of different mass than the computational ions.

Suggestion 2.4. "I found a statement of the article to be incomplete because the maximum starting temperature is not stated: pag. 5 "In the hottest case, we reduce the computational ion temperature by 102(5)...". In this context, the authors should write in the sentence the number of how many phonon excitations correspond to the condition to be the "hottest". Besides, I was wondering, is this "hottest" the "hottest" achievable because at higher temperatures the technique does not work any longer?"

We have modified the sentence "In the hottest **tested** case, we reduce the computational ion temperature **from 106(5) quanta to 3.9(2) quanta**" to include the word 'tested'. We expect this, along with the additional information in Section 3.5, will give the reader an understanding of what we have tested, and the physical limits of the protocol.

REVIEWERS' COMMENTS

Reviewer #1 (Remarks to the Author):

My comments have been addressed. I recommend publication.

Reviewer #2 (Remarks to the Author):

I have read the revised manuscript "Rapid Exchange Cooling with Trapped Ions" along with the authors' replies to all referee comments.

In my initial review, I raised a concern about whether this work's focus was applicable to the heating experienced in state-of-the-art QCCD devices. Based on the authors' response and included references, I now agree that the 10's-100+ quanta regime are still expected in even the best of devices. Given the satisfactory resolution of this primary concern, I reiterate that this is a very well-described protocol addressing a critical problem within QCCD architectures. I therefore recommend publication in Nature Communications.

Reviewer #3 (Remarks to the Author):

All the comments have been addressed in the new version of the manuscript. I consider the manuscript improved and complete. I recommend publication of the article.

Spencer Fallek
Response to Reviewers for 'Rapid Exchange Cooling with Trapped Ions'
Manuscript NCOMMS-23-44961

Dear Editor,

Thank you for your time and consideration. We have addressed editorial requests via the attached author checklist. There are no further comments to address from the reviewers.

Reviewer #1 (Remarks to the Author):

My comments have been addressed. I recommend publication.

Reviewer #2 (Remarks to the Author):

I have read the revised manuscript "Rapid Exchange Cooling with Trapped Ions" along with the authors' replies to all referee comments.

In my initial review, I raised a concern about whether this work's focus was applicable to the heating experienced in state-of-the-art QCCD devices. Based on the authors' response and included references, I now agree that the 10's-100+ quanta regime are still expected in even the best of devices. Given the satisfactory resolution of this primary concern, I reiterate that this is a very well-described protocol addressing a critical problem within QCCD architectures. I therefore recommend publication in Nature Communications.

Reviewer #3 (Remarks to the Author):

All the comments have been addressed in the new version of the manuscript. I consider the manuscript improved and complete. I recommend publication of the article.